# Assessment of ripple effect and spatial heterogeneity of total losses in the capital of China after a great catastrophe shocks

Zhengtao Zhang[1, 3], Ning Li[1, 2, 3], Wei Xie[4], Yu Liu[5], Jieling Feng[1, 3], Xi Chen[1, 3], Li Liu[1, 3]

[1]State Key Laboratory of Earth Surface Processes and Resource Ecology, Faculty of Geographical Science, Beijing Normal University, Beijing, 100875, China

[2]Key Laboratory of Environmental Change and Natural Disaster, MOE, Faculty of Geographical Science, Beijing Normal University, Beijing, 100875, China

[3]Academy of Disaster Reduction and Emergency Management, Ministry of Civil Affairs & Ministry of Education, Faculty of Geographical Science, Beijing, 100875, China

[4]China Center for Agricultural Policy, School of Advanced Agricultural Sciences, Peking University, No 5 Yiheyuan Road, Haidian District, Beijing, 100871, China

[5] Institutes of Science and Development, Chinese Academy of Sciences, No.15 Zhongguancun Beiyitiao, Haidian District, Beijing, 100190, China

*Correspondence to*: Ning Li (ningli@bnu.edu.cn)

**Abstract.** The total losses caused by natural disaster have spatial heterogeneity due to different economic development level inside the disaster-hit areas. This paper set the scenarios of direct economic loss to introduce the sectors' loss caused by 2008 Wenchuan earthquake into Beijing, utilized Adaptive Regional Input-Output (ARIO) model and Inter-regional ripple effect (IRRE) model. The purpose is to assess the ripple effects of indirect economic loss and spatial heterogeneity of both direct and indirect economic loss at the scale of smallest administrative divisions of China: streets/ (villages and towns). The results indicate that the district of Beijing with the most severe indirect economic loss is Chaoyang district; Finance & Insurance industry (#15) of Chaowai Street suffers the most in Chaoyang district, which is 1.46 times of its direct economic loss. During 2008-2014, the average annual GDP growth rate of Beijing could be decreased 3.63% affected by the catastrophe. Compared with the 8% of GDP growth rate target, the decreasing GDP growth rate is a significant and noticeable economic impact, and it can be efficiently reduced by increasing rescue effort and priority supporting the industries which are located in the seriously damaged regions.

## 1 Introduction

Economic losses caused by frequent natural disasters have been increased dramatically, and pose serious challenges to the world sustainable development and human safety(Munich, 2002). In April 2016, earthquakes with M_S>7 occurred within 8 days in succession in Japan(M_S7.3), Ecuador (M_S7.8), Burma(M_S7.2), and Afghanistan (M_S7.1), which caused large total economic losses to the above countries. The large economic losses caused by natural disasters should be assessed more accurately for improving the awareness of disaster impacts and the higher ability of disaster prevention and mitigation.

The Total economic losses caused by disasters usually incorporate *the direct economic loss* (Algermissen S.T., 1984;Lijun S, 1998;Jiren L, 2003;Ming T, 2014) and *the indirect economic loss.* For the direct economic loss, it belongs to the physical damage when it occurs in an instant and the induced physical impact after the disaster (Cochrane, 1997). For the indirect economic loss, it is generated by business interruption, imbalance between supply and demand, disorder between forward output and backward supply of sectors due to physical damage (Cochrane, 1997;FEMA, 2001). Its property of "invisible loss" makes it difficult to evaluate, but it cannot be ignored. Because indirect economic loss can increase, even outnumber direct economic loss along with the economic development, which underscore the significance of capturing the ripple effects accumulated along inter-regional and interindustrial linkages. The Input-Output (IO) model(Hallegatte et al., 2010;Wu et al., 2012;Hallegatte, 2014;Zhou et al., 2014;Xia et al., 2016) and Computable General Equilibrium (CGE) model(Rose and Liao, 2005;Guivarch et al., 2009;Xie et al., 2013;Rose, 2015) are two representative models which are commonly used to assess indirect economic loss.

However, most of the loss assessment methods only consider the overall loss value of the region (Xie et al., 2012), the losses in different regions inside the disaster-hit area have not yet been quantified. In modern city, the functional zoning have its own economic characteristics because of the industrial distribution, so there may be only one or two developed economic sectors existing in each functional zoning. If a catastrophe occurs, the direct economic losses of sectors will not evenly distributed inside the disaster-hit area due to the range of disaster impacts(Figure 1). In addition, as a result of the industrial linkage, the ripple effect of indirect economic loss can make a sector's loss spread to other related sectors in other region (e.g. the production reduction of an automobile manufacturing sector increases the unintended inventory (or decreases production )

of steel sector in other region, eventually making its indirect economic loss increasing)(Figure 1). Therefore, there are obvious spatial heterogeneity in direct and indirect economic losses of sectors inside the disaster-hit area. The distribution of total losses are inconsistent in two disaster-hit areas with the similar economic development degree (or whole total losses).

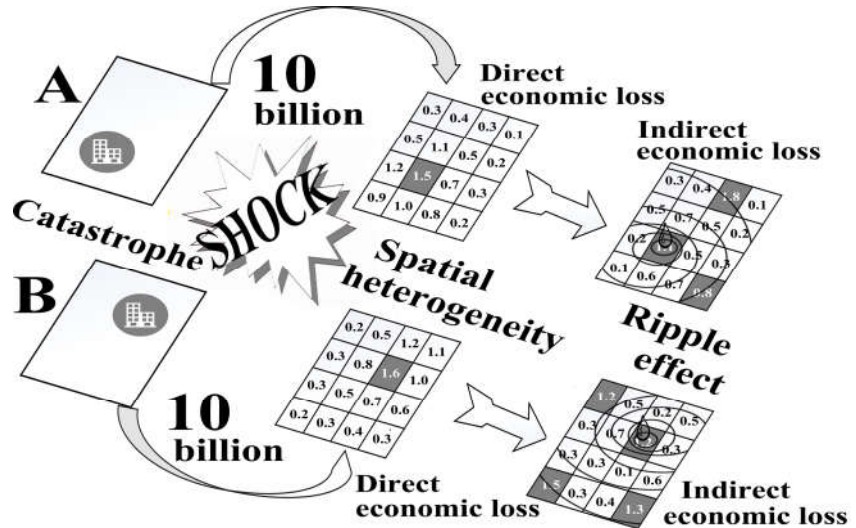

Figure 1. The diagrammatic drawing of spatial heterogeneity and ripple effect (Notes: A and B represent two disaster-hit areas with different functional zoning. If a catastrophe with the same intensity hits, both suffers DEL of USD 10 billion, but the losses inside areas are different due to functional zoning and disaster location. For ripple effect of IEL, the indirect economic losses is not only impacted by the distance of disaster occurring, but is impacted by inter-regional and interindustrial linkages)

Much attention has been paid to assessing the total loss of the whole disaster-hit area, but further study on refined assessment of direct economic loss and ripple effect of indirect economic loss inside the area is crucial to revealing the comprehensive disaster impacts, though internal economic data is difficult to acquire and the assessment models are rare. The results of the above research will play a significant role in different aspects. For governments, the policy maker can make more refined post-disaster recovery and reconstruction policies, and give the sector's priority to recovery with closer industrial linkage to make the economic system recover to pre-disaster level faster. For the insurance firms, they can add insurance categories to increase profits, also enhance the economic resilience of economic system. For the public, people can fully aware of the long-term impact of disaster and thus may change their economic activities (e.g. real estate investment).

2008 Wenchuan earthquake (2008 WCE) is designated as the catastrophe in this paper because it is the most destructive natural disasters since the founding of China in 1949. Beijing (BJ), one of the most important and developed metropolises in

China, is chosen to be the disaster-hit area due to its location which located at the North China seismic belt (Figure 2), and once occurred the earthquake (September 2nd 1679) with the same magnitude of 2008 WCE in the history. We introduce the sectors' direct economic losses caused by 2008 WCE into BJ through the reasonable assumption.

This paper is the first to assess and analyse the ripple effects of indirect economic loss and spatial heterogeneity of both direct and indirect economic loss inside the disaster-hit area at the scale of the smallest administrative divisions in China: streets (villages and towns) by means of the Adaptive Regional Input-Output (ARIO) model(Hallegatte, 2008) and Inter-regional ripple effect (IRRE) model.

This paper aims to decrease the long-term economic impacts caused by disasters and provide more comprehensive information for government to make better post-disaster recovery, disaster prevention and mitigation, rescue measures and funds allocation policies to the regions which may suffer seriously damage in the disaster aftermath.

## 2 Study area and Data

### 2.1 Study area

The 2008 Wenchuan earthquake (2008 WCE) is chosen in this paper as the catastrophe because it's an earthquake with the greatest magnitude and the highest damage degree since the founding of China in 1949, which affected 417 counties of 16 provinces and cities, with 69,226 people dead, and direct economic loss of USD 124 billion (the exchange rate of CNY to USD is 0.14 in 2008). According to all the affected provinces, Sichuan Province (SCP) suffered most serious disaster with the direct economic loss of USD 104.9 billion, accounting for 89% of the total losses (NCDR & MOST, 2008). The Wu's study(Wu et al., 2012) result shows that, indirect economic loss of SCP caused by the earthquake was about USD 42.1 billion, which was 40% of its direct economic loss. The potential impact of the disaster on indirect economic loss has aroused wide concern from both government and scholars (Rose et al., 1997;Okuyama, 2007;Li et al., 2013).

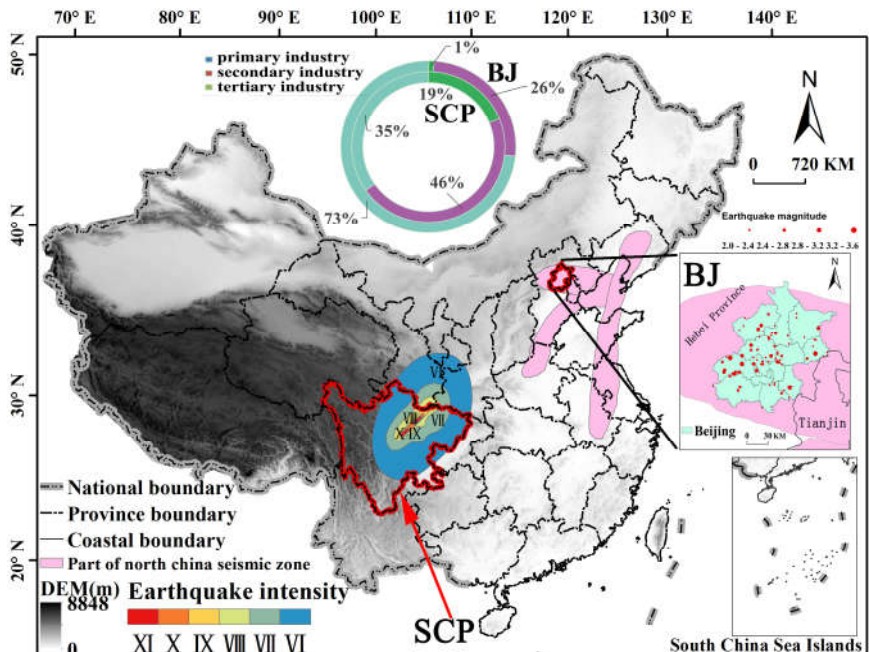

**Figure 2.** Sketch map of earthquakes in China, as well as the location of Beijing (BJ) and Sichuan province (SCP) (The top circle means the proportions of fixed asset stocks of primary, secondary and tertiary industry in BJ(outer ring) and SCP(inner ring). The pink area around BJ shows the north China earthquake zone; the size of red points on the map of BJ means the earthquake magnitudes and locations.)

Occurrence of an earthquake in a city may lead to more economic losses(Tantala et al., 2008;Hancilar et al., 2010). Therefore, we chose Beijing(BJ) as the disaster-hit area. Between 2002 and 2014, BJ occurred 74 earthquakes with the $M\_S>2$, mainly concentrated in Haidian District, Changping District and Shunyi District, as well as juncture between Mentougou District and Fangshan District(Figure 2).

**2.2 Data**

The data used in this study are mainly divided into three parts:

(1) Direct economic loss of BJ: The loss ratio of all sectors of SCP is taken as the scenario for direct economic loss of BJ. The loss of SCP comes from National Disaster Reduction Committee-Ministry of Science and Technology, with a total of 17 sectors (NCDR and MOST, 2008). The specific calculation method is shown in the section 3.1.

(2) Input-output table: 42 sectors of Input-output Tables of BJ and SCP are acquired from the Bureau of Statistics of BJ and SCP respectively. Since input-output table is published once every 5 years in China ( in the year ending with 2 and 7), and 2008 WCE occurred in the middle of 2008, so it is proper to use the latest 2007 input-output table to represent the industrial

linkage among sectors. In order to match 2007 input-output table with the statistics loss data of 17 sectors which comes from the National Commission for Disaster Reduction, the 42 sectors are reconsolidated into 17 sectors in accordance with the Industry Classification Standard of China.

(3) Number of enterprises, their total fixed assets and business incomes in 17 sectors within the smallest administrative divisions of streets/(villages and towns) of BJ: The data come from Beijing Macro Economic Social Development Basic Database. Because the time point of statistical data are not consistent, the administrative divisions of streets/(villages and towns) are not consistent with statistical data in the database, some streets/(villages and towns) are consolidated, e.g. Yizhuang Region (including national economic development zone), etc.

## 3 Research methods

### 3.1 Setting up Scenario of direct economic

The reason for setting this scenario is that there is no direct economic loss of 17 sectors of Beijing (BJ) due to no real disaster occurring. In order to solve this problem, The Scenario is set that: if an earthquake with the same intensity of 2008 Wenchuan earthquake (2008 WCE) occurs in BJ, and BJ has the same direct economic loss ratio of 17 sectors with that of Sichuan province (SCP). Before setting up Scenario, the economic conditions of two areas should be understood.

(1) Comparison of development between BJ and SCP

In terms of social and economic development degrees, SCP was categorized as a less developed region, its GDP was USD 175.1 billion (calculated based on 2008 constant price, the same below). BJ was categorized as a developed region, its GDP in 2008 was USD 146.8 billion, almost equivalent to the GDP of the whole province of SCP. Meanwhile, in 2008, per Capita GDP of BJ was 4.31times of SCP, the fixed asset stocks was 2.09 times. The per capita GDP of SCP only ranks 24[th] of the 34 provinces and cities of China, while that of BJ ranks the second, and is almost more than double as the per capita of Upper Middle Income country (5511 USD in 2008, from World Bank). The population density of SCP is 168/km$^2$, which ranks 25[th], and that of BJ is up to 1079/km$^2$, which ranks 4[th]. The proportion of the tertiary industry of BJ reached

73 percent. So, the developed secondary industry and tertiary industry as well as closer industrial linkage may lead to larger indirect economic loss is worthy of attention.

(2) Scenario of direct economic loss

The core thought of scenario of direct economic loss is that both areas have the same sector loss rates. The core formula of the scenario is calculated as followed:

$$DEL(BJ, n, 2008) = \frac{DEL(SCP, n, 2008)}{CAP(SCP, n, 2008)} \times CAP(BJ, n, 2008) \tag{1}$$

Where, $DEL$ (SCP (or BJ), $n$, 2008) is the direct economic losses of $n^{th}$ sector of SCP (or BJ) after catastrophe. $CAP$ (SCP (or BJ), $n$, 2008) is fixed asset stock of $n^{th}$ sector of SCP (or BJ) in 2008, The fixed assets stocks were calculated by the perpetual inventory method(Hall and Jones, 1999) according to the total investment in fixed assets of the whole society.

This method can visually reflect the shock of 2008 WCE on BJ under the scenario of same sector's loss ratio, and can reduce the calculation uncertainty of direct economic loss caused by different economic development degrees between two regions. The results are illustrated in the Table 1. It can be found that, economic pattern of BJ was mainly dominated by the tertiary industry, while SCP was dominated by the secondary industry, and it was in an industrial transformation period.

**Table 1.** Calculation Results of stock of fixed assets and direct economic loss of two areas

| ID | Sectors | Insdu--stry | Fixed asset stocks | | Loss Ratio | DEL | |
|----|---------|-------------|------|------|------|------|------|
| | | | SCP | BJ | | SCP | BJ |
| 1 | Agriculture | 1 | 491.7 | 13.93 | 0.03 | 16.80 | 0.48 |
| 2 | Mining Industry | 2 | 138.69 | 18.73 | 0.10 | 14.00 | 1.89 |
| 3 | Food Manufacturing & Tobacco process | 2 | 138.61 | 28.97 | 0.18 | 25.20 | 5.27 |
| 4 | Textile Manufacturing | 2 | 34.65 | 10.71 | 0.18 | 6.30 | 1.95 |
| 5 | Wood Processing & Furniture Manufacturing | 2 | 34.65 | 23.09 | 0.18 | 6.30 | 4.20 |
| 6 | Coke, Gas &Oil Process | 2 | 10.40 | 31.24 | 0.08 | 0.84 | 2.52 |
| 7 | Chemical Industry | 2 | 97.03 | 68.33 | 0.18 | 17.50 | 12.32 |
| 8 | Nonmetallic Mineral Manufacturing | 2 | 41.58 | 17.81 | 0.18 | 7.56 | 3.24 |
| 9 | Metallic Products Manufacturing | 2 | 103.96 | 52.06 | 0.18 | 18.20 | 9.11 |
| 10 | Mechanical Equipment Manufacturing | 2 | 173.27 | 263.85 | 0.18 | 30.80 | 46.90 |
| 11 | Electricity, Steam, Hot-water Production & Supply | 2 | 623.59 | 265.27 | 0.18 | 112.00 | 47.64 |

| 12 | Building Trade | 2 | 32.96 | 30.98 | 0.11 | 3.50 | 3.29 |
|----|----------------|---|-------|-------|------|------|------|
| 13 | Transportation, Post & Telecommunications | 3 | 708.67 | 872.57 | 0.17 | 117.60 | 144.80 |
| 14 | Commerce & Catering | 3 | 78.10 | 89.83 | 0.16 | 12.60 | 14.49 |
| 15 | Finance & Insurance | 3 | 1082.59 | 869.16 | 0.14 | 154.00 | 123.64 |
| 16 | Specific Service Management | 3 | 126.37 | 428.71 | 0.07 | 8.4.0 | 28.50 |
| 17 | Other Services | 3 | 425.80 | 562.73 | 0.18 | 77.00 | 101.76 |

*Note: (1) unit: USD 0.1 billion. Calculated based on 2008 constant price, the same below; (2) SCP means Sichuan province, BJ means Beijing, DEL means direct economic loss.

(3) Setting up damage intensity scenarios of BJ

It can be seen from Figure 2, scope of the areas with seismic intensity higher than IX degree covers all districts and counties of BJ. Furthermore, due to lack of real loss data of BJ, if we add the seismic source location and the different seismic intensity ranges, the simulation uncertainty will be greatly increased instead. Therefore, we assume that the intensity of disaster damage is evenly distributed in the space, and neither seismic source location nor intensity attenuation range is set. The upper limit of the losses caused by a violent earthquake in BJ is given through such extreme assumption.

### 3.2 Setting rescue scenes

After a catastrophe, Chinese government's rescue policy are "concentrating the whole country's efforts on rescue at all costs ", the rescue and reconstruction are led by the central government, which are not depend on the market (e.g. insurance). For recovery and reconstruction work of 2008 WCE, Chinese government provided 120% rescue efforts compare with pre-disaster production capacity(NCDR and MOST, 2008).With the consideration of importance of BJ and its economic development degree, if a devastating earthquake occurs, rescue efforts may reach 150%, and recovery period may be shorter. For this purpose, three rescue efforts scenes are set and they are illustrated in the Table 2. Price elastic coefficient parameter is 0.9 (Wu et al., 2012; Xie et al., 2012). By comparison of indirect economic losses of BJ under different rescue scene, the necessity of improving rescue efforts is evaluated.

Rescue scene A: Natural recovery. Only satisfying the demand on its basic disaster relief, just depending on BJ's own economic structure and industrial linkage to make the economic system recovered to its pre-disaster level.

Rescue scene B: BJ is provided with 120 % of rescue efforts compare with pre-disaster production capacity according to the real rescue policy for 2008 WCE. Scene B1 is the maximization of production capacity accomplished within 6 months, and scene B2 is to improve the production capacity to maximum 3 months ahead of schedule of Scene B1.

Rescue scene C: BJ is provided with stronger rescue efforts (150%) and shorter production capacity improvement period (3months), considering the economic development degree, position, and importance of BJ.

**Table 2.** ARIO model parameters and parameter setting of different post-disaster rescue scenes of Beijing (BJ)

| Parameter | Description | Scene A | Scene B | | Scene C |
| | | | Scene B1 | Scene B2 | |
|---|---|---|---|---|---|
| $a_b$ | Production capacity pre-disaster | 100% | 100% | 100% | 100% |
| $a_{max}$ | Maximum production capacity post-disaster | - | 120% | 120% | 150% |
| $\tau$ | Adaptation time | - | 6 month | 3 month | 3 month |
| $\xi$ | Price elastic coefficient | 0.9 | 0.9 | 0.9 | 0.9 |

### 3.3 The assessment model of indirect economic loss

The paper utilized the Adaptive Regional Input-Output(ARIO) Model to dynamically assess the indirect economic loss. ARIO model was proposed by Hallegatte, and it was successfully applied to the assessment of indirect economic loss caused by 2005 Hurricane Katrina(Hallegatte, 2008, 2014), flood in Mumbai, India,(Ranger et al., 2010)and 2008 WCE (Wu et al, 2012). This model is based on traditional IO model, and combines some advantages of CGE model. CGE model considers many complicated factors based on nonlinear thought such as market feedback, price change, but it also demand detailed statistical data and complicated process of parameter validation. These data are difficult to acquire at the scale of "unban" even "streets/ (villages and towns)". Therefore, ARIO model is appropriate in this study, it takes a full consideration on economic characteristics after the disaster, such as the change of production capacity of various economic sectors, production bottleneck, and capacity restriction caused by the impact of production reduction, production halts, and industrial linkage between various sectors , to make a dynamic simulation on the change of balance of supply and demand of the economic system in the time period from catastrophes occurrence to the completion of reconstruction, so as to describe the impact of the disaster to the

regional economy. The modeling time-step width is one month. Key parameters of the model (Table 2) and the core formula are illustrated as follows:

$$Y(j) = \overbrace{\sum_j A(i,j)Y(j)}^{intermediate\ demand} + \overbrace{LFD(i) + E(i) + HD(i) + \sum_j D(j,i)}^{final\ demand} \tag{2}$$

The parameter on the left side of the formula is the total output($Y(j)$), and parameters on the right side of the formula are the demand, including intermediate demand and final demand respectively. The cascading impact of disaster on economic system is reflected in: change of the demand (intermediate demand and $LFD$) within the disaster-hit areas and change of import ($E$) from the outside; increase of demands of households ($HD$) and commercial sectors ($D$); reduction ($D$) of production capacity caused by the reduction of capital, etc.

In order to better understand the ARIO model, some key parameters or processing step are explained: For the imports, the ARIO model copes with it from three aspects: i) improving the traditional IO table to the Local Input-Output (LIO) table based on the coefficient $P_j$, $P_j$= (*Production–Export*) / (*Production–Export +Import*). The model multiplies the values of quadrant *I* and quadrant *II* of traditional IO table by coefficient $P_j$ so as to remove the import part from the production and services of intermediate consumption/ inputs and final consumption; ii) setting substitutability of sectors: The model sets the substitutability of sectors based on the conditions that whether the sectors' productions and services can be imported from the outside of disaster-hit area. This study sets the manufacturing sectors (#3-#10), construction sector (#12) and Transportation, Post & Telecommunications (#13) cannot be substituted, other sectors' productions and services can be imported from the outside areas; iii) Imports changes during the reconstruction period: if a sectors can be substituted and its production capacity is insufficient, the import part is introduced. The imports amount depends on the relationship between demand and production, the parameter of import delay. The specific formula is as follows:

$$I(j) + \frac{TD^t(i) - Y(i)}{TD^t(i)} A(j,i) \frac{\Delta t}{t_A^{\downarrow}} \xrightarrow{\Delta t} I(j) \tag{3}$$

The sector $j$ is the sector who consumes the goods or services produced by sector $i$, $TD(i)$ and $Y(i)$ is the demand and production of sector $i$, $A$ is the intermediate consumption, $\Delta t$ is the time step of model, $t_A^{\downarrow}$ is the parameter of import delay,

and the initial $I(j)$ is the import of sector $j$ at $t^{th}$ month during reconstruction period, which is decided by the ratio of the production of $t^{th}$ month and the production of pre-disaster. The formula is shown as follows:

$$I(t+1, j) = I(t, j) \times \frac{Y(t+1, j)}{Y(1, j)}$$ (4)

All these steps are calculated based on the LIO table and the dynamic changes of model, therefore, the imports factor is an endogenous variable.

For the change in price, it's also an endogenous variable, calculated in this study by the relationship between demand and production in the disaster aftermath. The ARIO model assumes that commodity prices respond linearly to the production, and uses a single parameter of price inflation variable (parameter $\xi$ in Table 2) for the economic system in applying these prices. Therefore, the change in prices do not strongly feed back into the simulation results. However, The Chinese government will strictly control the changes in price, avoiding the serious inflation in the disaster aftermath. As a results that these assumptions accord with the characteristics of rescue system of Chinese government after the catastrophe. The price inflation variable is exogenous because it changes with the different economic systems of disaster-hit areas.

### 3.4 The Spatial diffusion (SDN) model of direct economic loss

Direct economic loss is mainly caused by the damage of fixed assets, belonging to inventory change. And as mentioned in the section 3.4, this study assumes the intensity of disaster damage is evenly distributed in the space of Beijing. Therefore, by utilizing the stock of fixed assets data of the streets/ (villages and towns) of BJ, post-disaster direct economic loss can be spatial extended into areas of 321 streets/(villages and towns). The core of formula of SDN model is as follows:

$$DEL_{str}(p) = \frac{CAP_{str}(p)}{CAP_{BJ}} \times DEL_{BJ} (p = 1, 2, 3 \cdots n)$$ (5)

Where $DEL_{str}(p)$ is direct economic loss of $p^{th}$ street of BJ, $CAP_{str}(p)$ stands for the stock of fixed assets of $p^{th}$ street, $DEL_{BJ}/CAP_{BJ}$ stands for direct economic loss/stock of fixed asset of BJ, and $n$ stands for 321 streets/ (villages and towns) of BJ. The stock of fixed asset is used in SDN model because that: i) the stock of fixed asset belongs to the concept of "stock", which is in line with the nature of direct economic loss. And the replacement costs of stock of fixed assets account for the

most in the statistical components of direct economic loss; ii) the core idea of the feedback of the production capacity of economic system in the ARIO model is that production capacities of sectors decreases x% when the stocks of fixed assets decreases x% caused by the disaster. Therefore, the using the sectors' stocks of fixed asset not only accords with the statistical significance of direct economic loss, but also the simulated mechanism of ARIO model.

### 3.5 The inter-regional ripple effect (IRRE) model of indirect economic loss

Indirect economic loss is mainly caused by the production reduction and industrial linkage of various economic sectors, belonging to flow change. Therefore, according to business income of sectors of streets/ (villages and towns), the ripple effects among sectors can be assessed at 321 streets/(villages and towns). The core of formula of IRRE model is shown as the following:

$$IEL_{str}(p,s,\alpha_{max}) = \frac{BIN_{str}(p,s)}{BIN_{BJ}(s)} \times IEL_{BJ}(s,\alpha_{max})(p=1,2,3\cdots n; \alpha_{max}=100\%,120\%,150\%) \tag{6}$$

Where, $IEL_{str}$ stands for indirect economic loss of streets/(villages and towns), $BIN_{str}$ stands for business income of streets/(villages and towns), $BIN_{BJ}/IEL_{BJ}$ stands for business income and indirect economic loss respectively, $p$ stands for $p^{th}$ street, $s$ stands for $s^{th}$ sector, $\alpha_{max}$ stands for three different rescue efforts scenes. The business income is used in IRRE model because that: i) the business income is the flow data, which is in line with concept of "flow" of indirect economic loss; ii) in the earthquake disaster aftermath, the production reduction due to the limitation of production capacity, production bottleneck, and industrial linkage can be reflected in the business income of the affected year. The business income of a sector will be greatly affected if that sector suffers large indirect economic loss. Therefore, it's reasonable to use business income in the IRRE model.

According to established IRRE model, the former studies that only obtain the direct and indirect economic loss of a whole region, without more specific losses information inside the region can be resolved. we can not only accurately spread the post-disaster losses from the whole city to districts-streets/(villages and towns), which not only helps to understand spatial heterogeneity of the losses, but also can propose "uneven" post-disaster recovery measures by analyzing the relationship between direct and indirect economic loss of sectors in various streets, to help the government find out optimal recovery and

reconstruction solutions for different areas by choosing either increasing rescue efforts to improve production capacity, or importing substitutes product from outside, so as to achieve a quicker recovery to the post-disaster level.

## 4 Results of loss assessment

### 4.1 Assessment results of overall losses under rescue scenes

According to the results of simulation, under an extreme scenario, Beijing (BJ) is likely to suffer the USD 150.6 billion (the exchange rate of CNY to USD is 0.14 in 2008) of total loss, including USD 88.1 billion of direct economic loss and USD 62.5 billion of indirect economic loss (Table 3).   The indirect economic loss accounts for 41.5% of total loss, and is almost the same as the indirect economic loss. The total loss has exceeded the 2.6% of BJ's GDP in 2008, and accounts for 3.6% of the national GDP of 2008. During the 6 years after the catastrophe occurred (2008-2014), the average annual GDP growth rate of

BJ could be decreased from 8.55% to 4.91%, the decreasing value was up to 3.63%, however, the goal for GDP growth rate of BJ was 8%, therefore the economic impact caused by the catastrophe seriously impeded the sustainable development in the future (Table 4).

**Table 3.** Evaluation results of indirect economic loss of Beijing (BJ) under different rescue scenes

| Rescue Scenes | DEL* (billion USD) | IEL* (billion USD) | DEL* /Total loss | IEL* /Total loss | IEL* /DEL* |
|---|---|---|---|---|---|
| Scene A | 88.1 | 62.5 | 58% | 42% | 71% |
| Scene B1 | 88.1 | 41.2 | 68% | 32% | 47% |
| Scene B2 | 88.1 | 37.1 | 70% | 30% | 42% |
| Scene C | 88.1 | 17.5 | 83% | 17% | 20% |

*Note: DEL means direct economic loss.

The Table 3 shows that, along with the intensifying rescue efforts, the indirect economic loss decreases gradually and thus the total losses decrease. When rescue efforts intensify from rescue scene A(natural recovery) to rescue scene C(150% of rescue efforts), the indirect economic loss is reduced by USD 45.1 billion, and the percentage of indirect economic loss to total losses is reduced from 42% to 17%. By comparing the value of "IEL/ DEL", it can be seen that, when BJ suffers USD 100 of direct economic loss, its indirect economic loss is reduced from USD 71 of rescue scene A to USD 20 of rescue scene C.

Where does the reduced USD 51 of indirect economic loss come from? We can figure it out by further analyzing the reasons: after disaster, damage of fixed assets leads to the inevitable outcome of reduction of production capacity, and the impact of industrial linkage on production capacity could be large or small. However, BJ is featured with strong industrial linkage, which brings difficult economic recovery, long recovery period, and slow capital accumulation. Under the same rescue efforts, when the time for improve production capacity to maximization is 3 months ahead of schedule (scene from B1 to B2), the indirect economic loss reduces by USD 5; under the same recovery period of 3 months, when rescue efforts increase by 30% (scene from B2 to C), the indirect economic loss will reduce by USD 22.

**Table 4.** The comparison of GDP growth rate between no catastrophe and assessment result under scene A six years after the catastrophe occurred

| GDP growth rate | 2009 | 2010 | 2011 | 2012 | 2013 | 2014 | average |
|---|---|---|---|---|---|---|---|
| No catastrophe | 10.2% | 10.3% | 8.1% | 7.7% | 7.7% | 7.3% | 8.55% |
| Catastrophe occurred | -1.1% | 4.3% | 5.3% | 6.5% | 7.3% | 7.2% | 4.92% |

In conclusion, rescue efforts play a more important role in the post-disaster recovery and reconstruction. It's necessary to study how to properly use and scientifically allocate rescue and recovery funds according to the losses of different regions and different economic sectors. Therefore, the study will conduct an in-depth comparative analysis of the ripple effects of indirect economic loss and spatial heterogeneity of direct economic loss in the next section.

## 4.2 Spatial heterogeneity of direct economic loss and ripple effects of indirect economic loss

The losses are closely associated with the economic development of various regions of BJ, so they are not evenly distributed. If rescue funds are evenly distributed according to the total losses without considering spatial differences of specific losses, it is likely to cause improper allocation of funds and reduce rescue efficiency, which is not conductive to post-disaster recovery and reconstruction.

(1) Differences of direct and indirect economic loss on the various districts

Statistics losses data on the spatial scales of districts/counties are based on those of streets/(villages and towns), which can be found in a macro perspective (Figure 3). The three administrative districts with the most serious direct economic loss are Haidian District (USD 21.4 billion), Xicheng District (USD 17.3 billion), and Chaoyang District (USD 12.9 billion), while the

three administrative districts with the most serious indirect economic loss are Chaoyang District (USD 9.8 billion), Haidian District (USD 8.6 billion), and Xicheng District (USD 6.4 billion). Among which, the indirect economic loss of Chaoyang District accounts for 76% of its direct economic loss, and the total economic losses is 1.76 times of its direct economic loss. Besides, although the indirect economic loss of Chaoyang District is less than that of Haidian District, its indirect economic loss is USD 1.2 billion higher than that of Haidian District due to its developed tertiary industry and strong industrial linkages, demonstrating that, indirect economic loss is relatively higher in economically developed regions, so it shall not be ignored in those more developed regions.

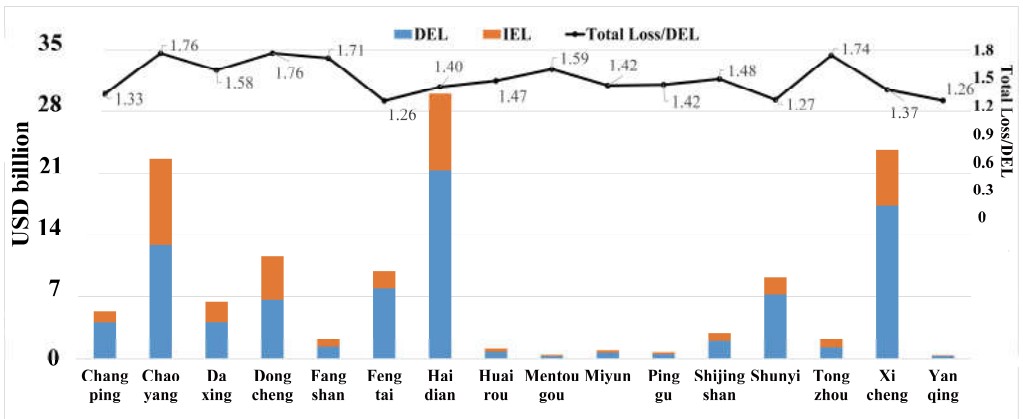

**Figure 3.** Direct and indirect economic loss of districts of Beijing (BJ), and ratios between total losses and direct economic loss (Notes: DEL means direct economic loss, and IEL means indirect economic loss )

In addition to the ratio of total losses and direct economic loss in Chaoyang District reaching 1.76, the ratios in Dongcheng District, Fangshan District, and Tongzhou District are also higher than 1.70. The total losses of Fangshan District and Tongzhou District are relatively less due to their relative poor economic development, but according to the current industry structure and development planning, the government shall pay high attention to the establishment of sufficient disaster prevention and mitigation plans as well as recovery and reconstruction strategies in case of occurrence of catastrophe in the future.

 (2) Spatial heterogeneity of direct and indirect economic loss

It can be seen from Figure 4, firstly, direct and indirect economic loss are mainly concentrated in main urban areas (Dongcheng District, Xicheng District, Haidian district, Chaoyang District, Shijingshan District, and Fengtai District). The

longer the distance to the main urban areas is, the smaller the losses will be. Secondly, according to spatial distribution, even though the indirect economic loss is smaller than direct economic loss, the indirect economic loss of some streets/ (villages and towns) are higher than their corresponding direct economic loss. Therefore, the result shows that even though the indirect economic loss of a region is smaller than direct economic loss, some places inside the region may greater than direct economic loss. The result can help the government confirm the important rescue places with more clearly. Thirdly, according to spatial agglomeration degree (Figure 4), the spatial aggregation of direct economic loss is 0.14 (Moran's index), and that of indirect economic loss is 0.27 (Moran's index), both are significant at the 99 percent significant level. Therefore, the spatial distribution of high value of indirect economic loss is more concentrated in the city center, while distribution with high direct economic loss value are relatively distributed dispersedly.

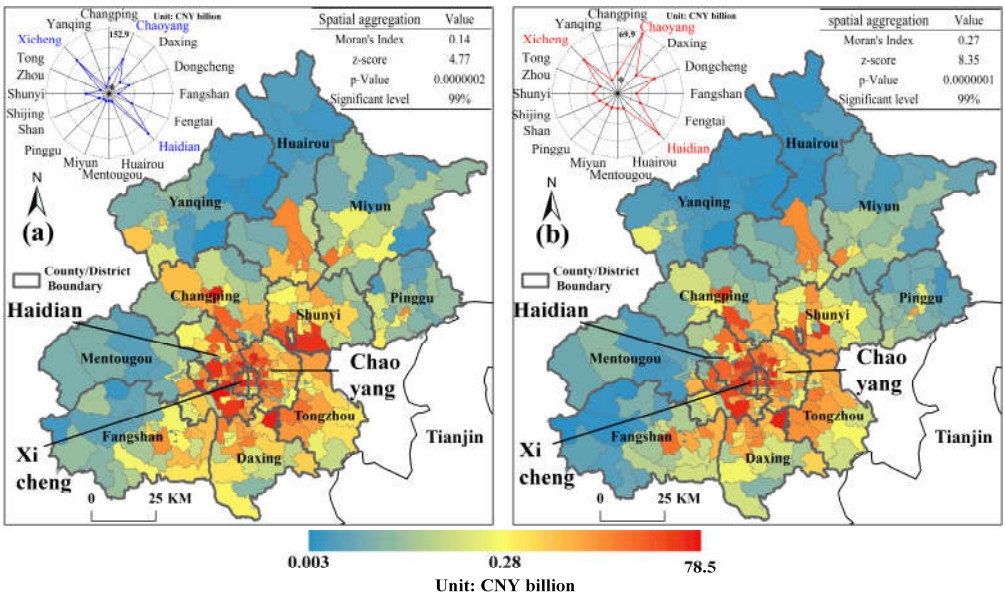

**Figure 4.** Spatial heterogeneity of direct economic loss (a) and ripple effects (b) of indirect economic loss (The rose diagram in the upper left-hand corner of Figure represents loss values of districts. The table in the upper right-hand corner of Figure represents the results of spatial aggregation calculated by the method of Spatial Autocorrelation (Moran's I), the global Moran's I is an index to measure spatial correlation, which is more obvious when its value is closer to 1. The z-score and p-Value are used to judge confidence coefficient level, the results show that both of spatial aggregation of DEL and IEL exceed the 99% confidence interval.)

The reasons may include: 1) Direct economic loss is mainly generated by the damage of fixed assets and according to the spatial structure planning of BJ of "Two-axis, two-belt, and multiple centers"(Song, 2009), most of traditional manufacture enterprises has been moved outside the city center, so the fixed assets values of peripheral districts are relatively high, and the post-disaster direct economic loss is higher in these districts; 2) BJ has a developed tertiary industry, which accounts for 77.9% of the total GDP of the city. Affected by the population and commercial distribution, the tertiary industry is mainly distributed in the city center, so the spatial distribution of the tertiary industry is mainly concentrated in the city center.

**4.3 Analysis of ripple effects of indirect economic loss and optimal recovery measures**

Chaoyang District suffers the highest indirect economic loss, so the analysis is made by taking Chaoyang District as an example. Areas with high value in Chaoyang District are mainly concentrated between the East 2nd Ring Road and the East 3rd Ring Road, and the three streets with the highest value are Chaowai Street (USD 1.6 billion), Jianwai Street (USD 1.5 billion), and Maizidian Street (USD 0.9 billion). Among them, the indirect economic loss of Chaowai Street is higher than its direct economic loss (USD 0.6 billion), and it is 2.6 times of its direct economic loss. According to the IRRE model, the reason for high indirect economic loss of Chaowai Street can be further analyzed through its various sectors. The indirect economic loss of Chaowai Street is mainly focused on the tertiary industry such as Commerce & Catering industry (#14) as well as Finance & Insurance industry (#15) (Figure 5). The indirect economic loss of Commerce & Catering industry (#14) is 39 times of its direct economic loss, accounts for 25% of total indirect economic loss of Chaowai Street. The indirect economic loss of Finance & Insurance industry (#15) of the street is 1.46 times of its direct economic loss, accounts for 68% of the losses of the street. Government should pay attention to these two sectors when making post-disaster recovery and reconstruction policy for Chaowai Street.

In terms of the optimal recovery measure for Commerce & Catering industry (#14), due to its small direct economic loss, its indirect economic loss is mainly caused by industrial linkage, so measure to increase market demands can be taken to pull the economic growth in order to speed up its recovery; In terms of optimal recovery measure for Finance & Insurance industry (#15), government shall consider the post-disaster recovery of the overall economic system. i) the severe damage of fixed assets of the secondary industry leads to the seriously inadequate intermediate input; ii) the decrease of the demands of

downstream industries leads to the decrease of production or the increase of inventory of the secondary industry may be the two major reasons to cause the increase of the indirect economic loss of the tertiary industry. When government makes the recovery and reconstruction policy, it is suggested not only to consider the impact of industrial linkage, but also to promote the post-disaster production capacity of the secondary industry, which can promote BJ's economic system recover to the

pre-disaster level. In addition, the industries distributed in Chaoyang district are mainly tertiary industry, so industrial structure of Chaoyang is also mainly dominated by the tertiary industry, once a great disaster occurs, the high loss value of this district will be mainly caused by the losses of the tertiary industry.

Besides, the manufacturing industry (automobile manufacturing) is mainly distributed in districts such as Shunyi District, and the industrial structure of this District is also mainly dominated by the secondary industry, so the majority of total losses in

Shunyi District are the losses of the secondary industry Therefore, industrial distribution and industrial structure are key elements for determining the size of indirect economic loss of a region. In the course of post-disaster recovery and reconstruction, government needs to take different recovery measures according to the industrial distributions and structures of different regions.

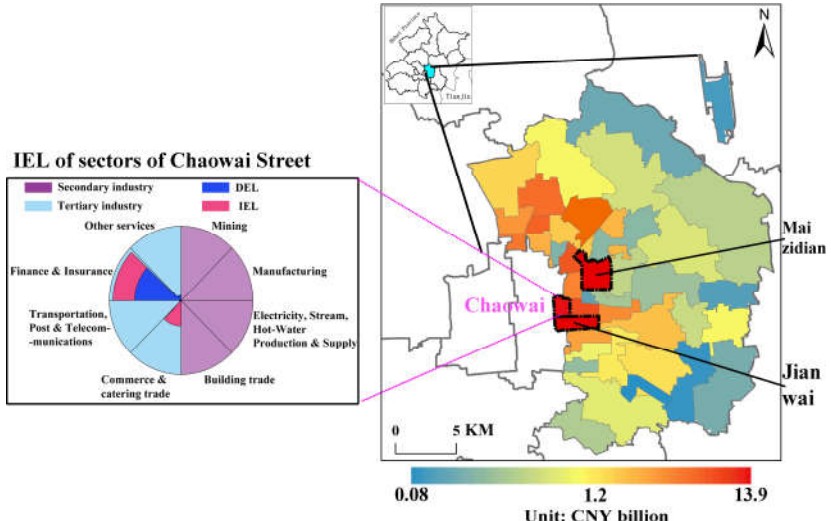

**Figure 5.** The right figure: the ripple effect of indirect economic loss (IEL) of Chaoyang District; The left figure: direct (DEL) and indirect (IEL) economic loss of sectors of the secondary industry and the tertiary industry of Chaowai Street

By analyzing ripple effects, the overall loss can be spatial extended into each street, and sectors' losses in each street can be further evaluated, which helps government policy-makers to intuitively get the information about the loss distribution, so as to properly allocate rescue efforts when taking recovery measures.

## 4.4 The ripple effects under different rescue scenes

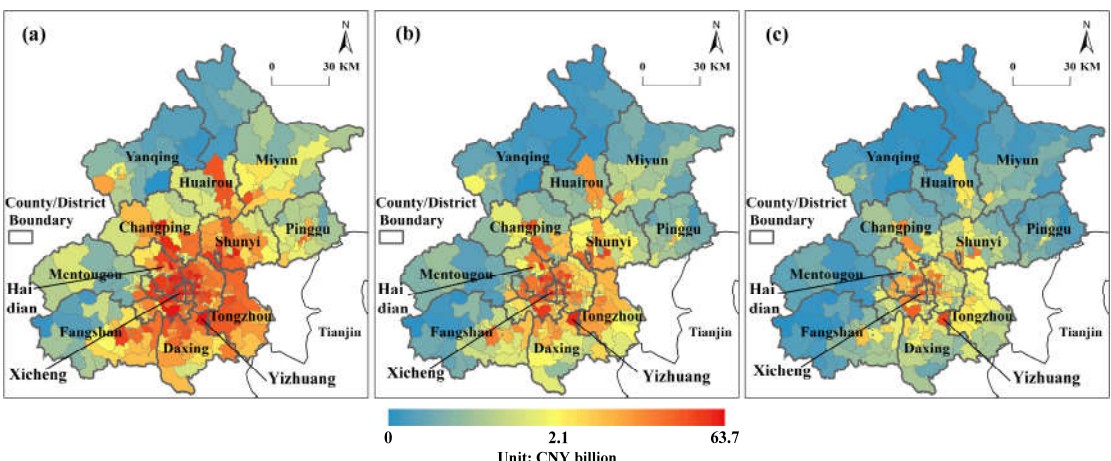

**Figure 6.** Evaluation results of indirect economic loss of streets/(villages and towns) of Beijing (BJ) under different rescue scenes (Figure (a): under scene A; Figure (b): under scene B1; Figure (c): under scene C)

According to the IRRE model, the spatial distribution of indirect economic loss under three post-disaster rescue scenes in BJ are illustrated in the Figure 6.

Under the rescue scene A of natural recovery, no rescue support will be given to the BJ's economic system, and recovery will be accomplished by relying on its own economic structure and industrial linkage, so its reconstruction and recovery period will be relatively longer with larger economic losses. It can be seen that the areas with high losses (Figure 6a) are mainly distributed in the city center of BJ, and the areas with the most severe losses are concentrated in the regions such as Xicheng District and Haidian District, and extended to the regions such as Changping District, Shunyi District, Tongzhou District, Fangshan District, etc. Under rescue effort scene B (scene B1: 120%, 6 months), due to the introduction of external recovery funds, and a series of recovery measures taken for economic system such as improving production capacity, substitution by increasing imports from outside, the overall loss is reduced. Spatially, indirect economic loss of various regions shows a decrease, and only some streets/ (villages and towns) see relatively high losses. The areas with high losses mainly concentrated

on city center areas, such as Xicheng District and Haidian District(Figure 6b). Under rescue effort scene C, the losses of each region are further reduced, but the spatial heterogeneity shows an obvious difference between high value loss and low value loss. (Figure 6c) The high value regions like Yizhuang region (including Economic Development Zone) still suffer losses without showing much decrease along with the improving rescue effort, while the losses of other regions reduce significantly.

Compared with rescue scene A, post-disaster affected areas* under rescue scene C is reduced by 4,364 km², accounting for 26.6% of the total area of BJ. It indicates that when the rescue effort is increased to 150% within 3 months, fixed assets of various economic sectors can be recovered with the fastest speed, and the industrial linkage is established rapidly, which will help the BJ's economic system faster recover and finish reconstruction.

*Notes:   Affected area calculation is based on disaster magnitude (Freckleton et al., 2012). According to the Geometrical Interval
spatial-based segmentation algorithm in GIS, the loss data of four scenes is categorized into five grades of catastrophe (USD 3.75 billion), disaster (USD 0.65 billion), medium disaster (USD 0.11 billion), small disaster (USD 0.019 billion), micro disaster (USD 0.003 billion). The regions involved in corresponding indirect economic loss of disasters with the grade higher than micro disaster is defined as the post-disaster affected areas.

**5 Conclusion**

In order to reduce the long-term economic impacts caused by disasters and accelerate the economic recovery in the disaster aftermath, this paper sets two scenarios and three models to calculate and analyse the spatial heterogeneity of direct economic loss and ripple effects of indirect economic loss at streets/(villages and towns) of BJ. We set the scenarios of direct economic loss to introduce the sectors' loss caused by 2008 WCE into BJ, set rescue scenes to assess the necessity of improving rescue efforts, adopted ARIO model to evaluate the indirect economic loss of BJ, established SDN model and
IRRE model to assess the direct and indirect economic loss of 321 streets/(villages and towns) of BJ. The results indicate that high value of indirect economic loss is more concentrated in the centre of BJ compared with that of direct economic loss. Finance & Insurance industry (#15) of Chaowai Street in Chaoyang district suffers the most serious indirect economic loss. During the six years after catastrophe occurred (2008-2014), the average annual GDP growth rate of BJ could be decreased from 8.55% to 4.91%, In terms of the 8% of BJ's GDP growth rate target, it's a significant and noticeable economic impact.

Before the disaster, attention shall be paid to the adjustment and optimization of industrial structure; In disaster

emergency rescue stage, attention shall be paid to the transportation of relief material allocation; during post-disaster recovery and reconstruction stage, attention shall be paid to give the priority to the development of the industries with high industrial linkage coefficient, so as to speed up the economic recovery. Therefore, adjustment of industrial structure and strengthen of industrial linkage can mitigate the impact caused by natural disasters. Besides, optimization of industrial structure, and close industrial linkage is of great significance in regional growth. In order to achieve sustainable development of a region, a balanced point between economic development and disaster mitigation must be found, and by taking it as a reference, a strategy for the optimization and adjustment of industrial structure shall be established. The results can provide the scientific and effective support to find the above "balanced point" according to increase rescue effort and to priority support the industries which are located in the seriously damaged regions.

*Acknowledgement.* The financial support provided by National Key Research and Development Program--Global Change and Mitigation Project: Global change risk of population and economic system: mechanism and assessment, NO. 2016YFA0602403; Beijing Municipal Natural Science Foundation, NO. 9172010; Major Program of National Natural Science Foundation of China No. 91325302; National Basic Program of China (973 Program), NO. 2012CB955402; National Key R&D Program (2016YFA0602604); National Natural Science Foundation of China (No. 41171401, No. 41101506); The Fundamental Research Funds for the Central Universities, NO. 310421101.

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
