# Peer review of "Assessment of ripple effect and spatial heterogeneity of total losses in the capital of China after a great catastrophe shocks"

_Natural Hazards and Earth System Sciences, 2016_

## Referee Comment (RC1) · Anonymous Referee #1 · 12 Dec 2016

This paper addresses a very interesting topic: the economic evaluation of the ripple effect and spatial heterogeneity after a catastrophe, with an application to earthquakes in the one of the most developed regions of China. The paper is well innovative and well written. It does a good job analyzing the ripple effect and spatial heterogeneity of total economic losses (especially indirect economic loss) by the established IRRE model. The results that the loss can be spatial extended into each street, and sectors' losses in each street can be further evaluated are both meaningful and useful.

I have a few comments:

i) Page 3, Line 3. Writing the names of the DEL and IEL in Figure 1 instead of acronyms would make it easier for the readers, especially in the introduction. ii) Page

6, Line16. You refer to Sichuan Province is a less developed region in China, (Page 2, Line18) refer to Beijing is a developed metropolises in China. . .What is the criterion to judge their economic development degreeïij§ iii) Page10, Line10. The SDN model, "DELBJ/CAPBJ stands for direct economic loss/stock of fixed asset of BJ"; Page10, Line18. The IRRE model, "BINstr stands for business income of streets/(villages and towns),". Why do you use stock of fixed asset to spread direct economic loss, use business income to spread indirect economic loss? iv) Page 10, Line16. What's the meaning of the parameters of spatial aggregation in Figure 4? You should illustr

Please also note the supplement to this comment:
http://www.nat-hazards-earth-syst-sci-discuss.net/nhess-2016-354/nhess-2016-354-RC1-supplement.pdf

---

## Short Comment (SC1) · 3 Jan 2017

This paper tries to assess the ripple effect and spatial distribution of total capital loss after a natural disaster event. Such topic is really interesting and important for natural disaster risk analysis. The data of capital loss for some specific sectors or regions are not easily to obtain, but such data are significant for the total economic impact analysis after the disaster. This paper provides a useful method to calculate the direct and indirect capital loss for each sector and also gives different scenarios to analyse the ripple effects of indirect economic loss.

However, there are still some places need to be clarified in this paper.

[Figure]

1. Page 2, line 7-10 Indirect economic loss has a very clear definition, but what is the direct part? It is better to give the definition of "direct economic loss".

2. In Page 2, line 13-16, it mentions, "The Input-Output (IO) model and Computable General Equilibrium (CGE) model are two representative models which are commonly used to assess indirect economic loss." but in this research, only the Input-Output model is used. Why? It is better to explain why you choose IO model, but not CGE model.

---

## Referee Comment (RC2) · Anonymous Referee #2 · 16 Jan 2017

The paper conducts a hypothetical case by applying 2008 Wenchuan earthquake into Beijing. It integrates scenario analysis, Adaptive Regional Input-Output (ARIO) model and Inter-regional ripple effect (IRRE) model to assess both direct and indirect economic loss of this hypothetical earthquake in Beijing. Specifically, the proposed IRRE model allows to investigate the spatial heterogeneity of direct and indirect loss. The paper appears to be one of the few papers conducting scenario analysis based on real past catastrophe and applying it onto another region. In this respect, the paper has certain level of novelty and it provides strong evidence to support pre-disaster preparation by Chinese local governments.

However, the paper is subject to a couple of limitations as follows: The paper utilizes

an ARIO model to simulate the post-catastrophe economy in Beijing. However, some key parameters in the ARIO model requires more detailed explanation. For adaptation mechanisms, apart from overproduction capacity, how did you deal with imports? Price/quantity changes after floods? It's fine not to consider those factors as they can be very complicated, but the authors need to clearly state those parameters. Is those factors considered exogenously or endogenously?

The paper distributes the direct and indirect economic loss into specific districts based on street-total capital ratio and street-total business income ratio, respectively. There are assumptions underlying here. Please state clearly about this.

The paper requires thorough language-editing, especially in section 4-Results of loss assessment. Some of the results are given mistakenly in 4.1, such as 'The indirect economic loss accounts for 41.5% of total loss, and almost the same as the indirect economic loss', 'The total loss has exceeded the 2.6% of BJ's GDP in 2008, and accounted for 3.6% of the national GDP of 2008.'

---

## Author Response (AR2)

**Responses to the reviewer's comments on the manuscript**

**"Assessment of ripple effect and spatial heterogeneity of total losses in the capital of China after a great catastrophe shocks"**

The authors would like to thank the reviewer for your efforts on this manuscript and providing us with insightful comments and suggestions to improve the quality of this manuscript. The following responses have been prepared to address reviewers' comments in a point-by-point fashion. And the sentences in red are the corresponding revised parts in our revised manuscript. The information of Page/Line in blue refers to the revised manuscript rather than the complete manuscript. We also attach a marked-up manuscript version in the below of the responses.

**Firstly, we have confirmed that the names for all authors on the title page are used as full first name.**

**Referee #1 Comments:**

*This paper addresses a very interesting topic: the economic evaluation of the ripple effect and spatial heterogeneity after a catastrophe, with an application to earthquakes in the one of the most developed regions of China. The paper is well innovative and well written. It does a good job analyzing the ripple effect and spatial heterogeneity of total economic losses (especially indirect economic loss) by the established IRRE model. The results that the loss can be spatial extended into each street, and sectors' losses in each street can be further evaluated are both meaningful and useful.*

*i) Page 3, Line 3. Writing the names of the DEL and IEL in Figure 1 instead of acronyms would make it easier for the readers, especially in the introduction.*

*Response:* **Thanks for your comments.** We have already changed the full names of the direct economic loss and indirect economic loss instead of acronyms "DEL" and "IEL" in Figure 1. And we also delete the words below the Figure 1: ", DEL means direct economic loss, IEL means indirect economic loss".

Please Check the Page 3, Line 7 in the revised manuscript.

*ii) Page 6, Line16. You refer to Sichuan Province is a less developed region in China, (Page 2, Line18) refer to Beijing is a developed metropolises in China…What is the criterion to judge their economic development degree?*

*Response:* **Thanks for your suggestion.** We added a paragraph on the Page 7, Line 4 to illustrate the development level between Sichuan province and Beijing. Besides, we also compare per captia of BJ with the world standard to highlight its development:

"The per capita GDP of SCP only ranks $24^{th}$ of the 34 provinces and cities of China, while that of BJ ranks the second, and is almost more than double as the per capita of upper middle income country (5511 USD in 2008, from World Bank). The population density of SCP is $168/km^2$, which ranks $25^{th}$, and that of BJ is up to $1079/km^2$, which ranks $4^{th}$."

*iii) Page10, Line10. The SDN model, "DELBJ/CAPBJ stands for direct economic loss/stock of fixed asset of BJ"; Page10, Line18. The IRRE model, "BINstr stands for business income of streets/(villages and towns),". Why do you use stock of fixed asset to spread direct economic loss, use business income to spread indirect economic loss?*

*Response:* **Thanks for your comment.**

The reasons that the stock of fixed asset is used to spread direct economic loss are: i) the direct economic loss belongs to the concept of "stock", the stock of fixed asset is in line with the nature of direct economic loss; ii) the destruction of sectors caused by earthquake disaster is mainly concentrated in destruction of stock of fixed asset, and the replacement costs of stock of fixed assets account for the most in the statistical components of direct economic loss; iii) the production capacity in the ARIO model is the key variable which is used to indicate the ability of increasing supply level to meet the post-disaster demand. The production capacity is set that it decreases x% when the stock of fixed asset decreases x% caused by the disaster. Therefore, using the sectors' stocks of fixed asset of streets/ (villages and towns) to spread direct economic loss not only accords with the statistical significance of direct economic loss, but also accords with the simulated mechanism of ARIO model.

The reasons that the business income is used to spread indirect economic loss are: i) the indirect economic loss

belongs to the concept of "flow", the business income is the flow data, so it's in line with the nature of indirect economic loss; ii) in the earthquake disaster aftermath, the enterprises will reduce production due to the limitation of production capacity, production bottleneck, and industrial linkage. All of the limitations can be reflected in the business income of the affected year.

The business income of a sector will be greatly affected if that sector suffers large indirect economic loss; iii) Due to the fact of data limitations, only business income meets the data requirement to spread the indirect economic loss in all the available statistics of streets/(villages and towns). In fact, the most accurate method to assess the indirect economic loss of every street/(village and town) in Beijing is to simulate them one by one based on ARIO model and their direct economic losses. However, it's impossible in China, because there are no Input-Output Table of every street/(village and town) and the import and export data between different regions. Therefore, according to the existing data, using business income data to spread indirect economic loss accords with the theoretical and practical significance.

We added a paragraph to describe the explanation.

On Page 12, Line 5: "The stock of fixed asset is used in SDN model because that: i) the stock of fixed asset belongs to the concept of "stock", which is in line with the nature of direct economic loss. And the replacement costs of stock of fixed assets account for the most in the statistical components of direct economic loss; ii) the core idea of the feedback of the production capacity of economic system in the ARIO model is that production capacities of sectors decreases x% when the stocks of fixed assets decreases x% caused by the disaster. Therefore, the using the sectors' stocks of fixed asset not only accords with the statistical significance of direct economic loss, but also the simulated mechanism of ARIO model."

On Page 12, Line 20: "The business income is used in IRRE model because that: i) the business income is the flow data, which is in line with concept of "flow" of indirect economic loss; ii) in the earthquake disaster aftermath, the production reduction due to the limitation of production capacity, production bottleneck, and industrial linkage can be reflected in the business income of the affected year. The business income of a sector will be greatly affected if that sector suffers large indirect economic loss. Therefore, it's reasonable to use business income in the IRRE model."

*iv) Page 10, Line16. What's the meaning of the parameters of spatial aggregation in Figure 4? You should illustrate them.*

*Response:* **We are agree with your suggestion.**

We have already added the corresponding illustration below the Figure 4 due to the space limitation. Please check the Page 17, Line 4 in the revised manuscript:

"the global Moran's I is an index to measure spatial correlation, which is more obvious when its value is closer to 1. The z-score and p-Value are used to judge confidence coefficient level, the results show that both of spatial aggregation of DEL and IEL exceed the 99% confidence interval."

**Referee #2 Comments:**

*The paper conducts a hypothetical case by applying 2008 Wenchuan earthquake into Beijing. It integrates scenario analysis, Adaptive Regional Input-Output (ARIO) model and Inter-regional ripple effect (IRRE) model to assess both direct and indirect economic loss of this hypothetical earthquake in Beijing. Specifically, the proposed IRRE model allows to investigate the spatial heterogeneity of direct and indirect loss. The paper appears to be one of the few papers conducting scenario analysis based on real past catastrophe and applying it onto another region. In this respect, the paper has certain level of novelty and it provides strong evidence to support pre-disaster preparation by Chinese local governments.*

*However, the paper is subject to a couple of limitations as follows:*

*i) The paper utilizes an ARIO model to simulate the post-catastrophe economy in Beijing. However, some key parameters in the ARIO model requires more detailed explanation. For adaptation mechanisms, apart from overproduction capacity, how did you deal with imports? Price/quantity changes after floods? It's fine not to consider those factors as they can be very complicated, but the authors need to clearly state those parameters. Is those factors considered exogenously or endogenously?*

*Response:* **Thanks for your comments.** The comment includes three questions, so we intend to answer them

separately.

*1. For adaptation mechanisms, apart from overproduction capacity, how did you deal with imports?*

*Response:* this paper copes with the imports from three aspects: i) Input-Output (IO) Table: The ARIO model improves the traditional IO Table in such way: For the every production *j*, there is coefficient $P_j$, $P_j$= (*Production–Export*) / (*Production–Export +Import*). The model multiplies the values of quadrant I and quadrant II of traditional IO table by coefficient $P_j$, creating a Local Input- Output table (LIO). The purpose is to remove the import part from the production and services of intermediate consumption/ inputs and final consumption, and to distinguish between sectors that produce goods and services that can be imported and those that produce goods and services that have to be locally produced. ii) setting substitutability of sectors: The model sets the substitutability of sectors based on the conditions that whether the sectors' productions and services can be imported from the outside of disaster-hit area. This study sets the manufacturing sectors (#3-#10), construction sector (#12) and Transportation, Post & Telecommunications (#13) cannot be substituted, other sectors' productions and services can be imported from the outside areas; iii) if a sectors can be substituted and its production capacity is insufficient during the reconstruction period, the import part is introduced. The imports amount depends on the relationship between demand and production, the parameter of import delay. The specific formula is as follows:

$$I(j)+\frac{TD^t(i)-Y(i)}{TD^t(i)}A(j,i)\frac{\Delta t}{t_A^{\downarrow}}\xrightarrow{\Delta t}I(j)$$

The sector j is the sector who consumes the goods or services produced by sector *i*, *TD(i)* and *Y(i)* is the demand and production of sector *i*, A is the intermediate consumption, *Δt* is the time step of model, $t_A^{\downarrow}$ is the parameter of import delay, and the initial *I(j)* is the import of sector *j* at $t^{th}$ month during reconstruction period, which is decided by the ratio of the production of $t^{th}$ month and the production of pre-disaster. The formula is shown as follows:

$$I(t+1, j) = I(t, j)\times\frac{Y(t+1, j)}{Y(1, j)}$$

*2. Price/quantity changes after floods?*

*Response:* we think the floods the referee proposed is actually the earthquake.

The Change in price in this study is calculated by the relationship between demand and production in the disaster aftermath. The ARIO model assumes that commodity prices respond linearly to the level of underproduction, and uses a single parameter of price inflation (parameter $\xi$ in Table 2) for the whole economy in applying these prices. Therefore, the changes in prices do not strongly feed back into the simulation results. But these assumptions accord with the rescue policy of Chinese government after the catastrophe. The government will strictly control the changes in price, avoiding the serious inflation in the disaster aftermath.

*3. Is those factors considered exogenously or endogenously?*

*Response:* According to the above description, the imports and prices are endogenously and the parameter of price inflation is exogenously. For the imports: the process for the imports data is based on the improved traditional IO table and the model itself; for the changes in prices, it's decided by the changes of demand and production. So the imports and prices are endogenously. For the price inflation, it depends on the condition of China's rescue policy. It changes with the different economic systems of disaster-hit areas. Therefore, it is exogenously.

We answer the above three questions together in the manuscript, please check the Page 10, Line 15 in the revised manuscript.

*ii) The paper distributes the direct and indirect economic loss into specific districts based on street-total capital ratio and street-total business income ratio, respectively. There are assumptions underlying here. Please state clearly about this.*

*Response:* **Thanks for your suggestion.**

As is mentioned in the section 3.1, according to the core damage scope of 2008 Wenchuan earthquake and the uncertainty of seismic source location and the seismic intensity ranges, this study assumes the intensity of disaster damage is evenly distributed in the space of Beijing, gives the upper limit of the losses caused by the catastrophe.

Based on the above assumption, the direct economic loss and indirect economic loss can be spread by the stock of

fixed asset and the business income respectively. For the distribution of direct economic loss, the stock of fixed asset data are used in the SDN model. Except for the reason mentioned in the section 3.4 that the stock fixed asset belongs to the inventory change, there are two more reasons: i) the replacement costs of stock of fixed assets account for the most in the statistical components of direct economic loss; ii) the core idea of the feedback of the production capacity of economic system in the ARIO model is that production capacities of sectors decreases x% when the stocks of fixed assets decreases x% caused by the disaster. Therefore, the using the sectors' stocks of fixed asset not only accords with the statistical significance of direct economic loss, but also the simulated mechanism of ARIO model.

For the distribution of indirect economic loss, the business income data are used in the IRRE model. Except for the reason mentioned in the section 3.5 that the business income is the flow data which is in line with the concept of "flow" of indirect economic loss, there is one more reason: in the earthquake disaster aftermath, the production reduction due to the limitation of production capacity, production bottleneck, and industrial linkage can be reflected in the business income of the affected year. The business income of a sector will be greatly affected if that sector suffers large indirect economic loss. Therefore, it's reasonable to use business income in the IRRE model.

The part of this comment is similar to the third comment of RC1, so we answer these two comments together in the revised manuscript. The corresponding description is shown on the Page 11, Line 18, Page 12, Line 5 and Page 12, Line 20.

*iii) The paper requires thorough language-editing, especially in section 4-Results of loss assessment. Some of the results are given mistakenly in 4.1, such as 'The indirect economic loss accounts for 41.5% of total loss, and almost the same as the indirect economic loss', 'The total loss has exceeded the 2.6% of BJ's GDP in 2008, and accounted for 3.6% of the national GDP of 2008.'*

*Response:* **Thanks for your comment.** We have already corrected the sentences and checked the section 4 again.

"The indirect economic loss accounts for 41.5% of total loss, and almost the same as the indirect economic loss." on the Page 13, Line 16 is changed as "The indirect economic loss accounts for 41.5% of total loss, and is almost the same as the indirect economic loss."

"The total loss has exceeded the 2.6% of BJ's GDP in 2008, and accounted for 3.6% of the national GDP of 2008."on Page 13, Line 18 is changed as "The total loss has exceeded the 2.6% of BJ's GDP in 2008, and accounts for 3.6% of the national GDP of 2008.".

"…the decreasing value is up to 3.63%,…" on Page 13, Line 20 is changed as "the decreasing value was up to 3.63%,".

"…therefore the economic impact caused by the catastrophe can seriously impede the sustainable development in the future (Table 4)" on Page 13, Line 21 is changed as "…therefore the economic impact caused by the catastrophe seriously impeded the sustainable development in the future (Table 4)".

The term "100 CNY" on Page 14, Line 5 for example is changed as "USD 100".

On Page 15, Line 7: "Statistics losses data on the spatial scales of districts/counties are based on those of streets/(villages and towns),", "are" is added in the sentence.

On Page 15, Line 12: "and its the total economic losses is 1.76 times of its direct economic loss.", "its" is changed as "the".

"BJ has a developed tertiary industry, accounting for 77.9% of the total GDP of the city, and affected by the population and commercial distribution," on Page 17, Line 10 is changed as "BJ has a developed tertiary industry, which accounts for 77.9% of the total GDP of the city. Affected by the population and commercial distribution,"

On Page 18, Line 7: "accounting for 25% of total indirect economic loss of Chaowai Street." is changed as "accounts for 25% of total indirect economic loss of Chaowai Street.". On Page 18, Line 8: "accounting for 68% of the losses of the street." is also changed as "accounts for 68% of the losses of the street."

On Page 18, Line 9: "Government departments should pay attention to …", the term "departments" is deleted as "Government should pay attention to …"

"Due to the severe damage of fixed assets of the secondary industry, seriously inadequate intermediate input, decrease of the demands of downstream industries, and decrease of production and increase of inventory of the secondary industry may be one of the major reasons to cause the increase of the indirect economic loss of the tertiary industry." on Page 18, Line 14 is changed as "i) the severe damage of fixed assets of the secondary industry leads to the

seriously inadequate intermediate input, ii) the decrease of the demands of downstream industries leads to the decrease of production or the increase of inventory of the secondary industry may be the two major reasons to cause the increase of the indirect economic loss of the tertiary industry."

On Page 18, Line 18: "When government making the recovery and reconstruction policy, it is suggested to not only consider the impact of industrial linkage, but also need to promote the post-disaster production capacity of the secondary industry," is changed as "When government makes the recovery and reconstruction policy, it is suggested not only to consider the impact of industrial linkage, but also to promote the post-disaster production capacity of the secondary industry,"

Some words in the sentences "their industrial structure is also mainly dominated by the secondary industry, so the majority of total loss in this district are the losses of the secondary industry Therefore, industrial distribution and industrial structure are key elements for determining the size of indirect economic loss of a region. In the course of post-disaster recovery and reconstruction, government departments need to …" on the Page 18, Line 24 is changed as "the industrial structure of this District is also mainly dominated by the secondary industry, so the majority of total losses in Shunyi District are the losses of the secondary industry. Therefore, industrial distribution and industrial structure are key elements for determining the size of indirect economic loss of a region. In the course of post-disaster recovery and reconstruction, government needs to …"

On Page 19, Line 5: "The left rigure: direct (DEL) and indirect (IEL) economic loss of sectors of the secondary industry and the tertiary industry of Chaowai Street" is changed as "The left figure: direct (DEL) and indirect (IEL) economic loss of sectors of the secondary industry and the tertiary industry of Chaowai Street".

On Page 20, Line 3: "(Figure (a): under scene A; Figure (b): under scene B1; Figure (c): under scene C)" is added in the below of Figure 6 for better understanding.

"…extend to the regions such as Changping District…"on the Page 20, Line 11 is changed as "…extended to the regions such as Changping District…"

"…still have losses without showing much decrease along with the rescue effort improving," on the Page 20, Line 18 is changed as "…still suffer losses without showing much decrease along with the improving rescue effort,"

"under rescue scene C" is added on the Page 21, Line 2 "post-disaster affected areas* under rescue scene C is reduced by 4,364 km2," for better understanding.

On Page 21, Line 3: "It indicates that when the rescue effort is increased to 150% 3 months ahead of schedule," is changed as "It indicates that when the rescue effort is increased to 150% within 3 months,"

**Short Comments:**

*This paper tries to assess the ripple effect and spatial distribution of total capital loss after a natural disaster event. Such topic is really interesting and important for natural disaster risk analysis. The data of capital loss for some specific sectors or regions are not easily to obtain, but such data are significant for the total economic impact analysis*

10   *after the disaster. This paper provides a useful method to calculate the direct and indirect capital loss for each sector and also gives different scenarios to analyse the ripple effects of indirect economic loss.*

*i) Page 2, line 7-10 Indirect economic loss has a very clear definition, but what is the direct part? It is better to give the definition of "direct economic loss".*

15   *Response:* **Thanks for your comments.** We agree with your comment. We have added the corresponding description on Page 2, Line 8, please check the revised manuscript.

"For the direct economic loss, it belongs to the physical damage when it occurs in an instant and the induced physical impact after the disaster (Cochrane, 1997)."

20   *ii) In Page 2, line 13-16, it mentions, "The Input-Output (IO) model and Computable General Equilibrium (CGE) model are two representative models which are commonly used to assess indirect economic loss." but in this research, only the Input-Output model is used. Why? It is better to explain why you choose IO model, but not CGE model.*

*Response:* **Thanks for your suggestion.** We added a paragraph to describe the difference between the IO model and CGE model on Page 9, Line 17:

25   "……This model is based on traditional IO model, and combines some advantages of CGE model. CGE model

considers many complicated factors based on nonlinear thought such as market feedback, price change, but it also demand detailed statistical data and complicated process of parameter validation. These data are difficult to acquire at the scale of "unban" even "streets/ (villages and towns)". Therefore, ARIO model is appropriate in this study, it takes a full consideration on economic characteristics……."

In addition, we also added two new financial supports fund: "Beijing Municipal Natural Science Foundation, NO. 9172010" and "Major Program of National Natural Science Foundation of China No. 91325302", on Page 22, Line 9, because they gave us much help during the modification period.

On Page 1, Line 4, 6, 8: the name of Institute is changed during the modification period.

10  On Page 1, Line12: the address of Number 5 (Yu Liu) is corrected as "Institutes of Science and Development, Chinese Academy of Sciences, No.15 Zhongguancun Beiyitiao, Haidian District, Beijing, 100190, China". Because the name of the Institute has changed during the period of review.

**Table** List of all relevant changes made in the revised manuscript

[revised manuscript text omitted]